# Ultra-Low-Power Wide Range Backscatter Communication Using Cellular Generated Carrier [note 1]

**DOI:** 10.3390/s21082663

**Published:** 2021-04-10

**Authors:** Muhammad Usman Sheikh, Boxuan Xie, Kalle Ruttik, Hüseyin Yiğitler, Riku Jäntti, Jyri Hämäläinen

**Affiliations:** Department of Communications and Networking, Aalto University, 02150 Espoo, Finland; boxuan.xie@aalto.fi (B.X.); kalle.ruttik@aalto.fi (K.R.); huseyin.yigitler@aalto.fi (H.Y.); riku.jantti@aalto.fi (R.J.); jyri.hamalainen@aalto.fi (J.H.)

**Keywords:** backscatter communication, LoRa backscatter, IoT, NB-IoT, BLE, ray tracing, low power wide area network, LPWAN, simulations, 3GPP, macrocellular, smart city

## Abstract

With the popularization of Internet-of-things (IoT) and wireless communication systems, a diverse set of applications in smart cities are emerging to improve the city-life. These applications usually require a large coverage area and minimal operation and maintenance cost. To this end, the recently emerging backscatter communication (BC) is gaining interest in both industry and academia as a new communication paradigm that provides high energy efficient communications that may even work in a battery-less mode and, thus, it is well suited for smart city applications. However, the coverage of BC in urban area deployments is not available, and the feasibility of its utilization for smart city applications is not known. In this article, we present a comprehensive coverage study of a practical cellular carrier-based BC system for indoor and outdoor scenarios in a downtown area of a Helsinki city. In particular, we evaluate the coverage outage performance of different low-power and wide area technologies, i.e., long range (LoRa) backscatter, arrow band-Internet of Things (NB-IoT), and Bluetooth low energy (BLE) based BC at different frequencies of operation. To do so, we carry out a comprehensive campaign of simulations while using a sophisticated three-dimensional (3D) ray tracing (RT) tool, ITU outdoor model, and 3rd generation partnership project (3GPP) indoor hotspot model. This study also covers the energy harvesting aspects of backscatter device, and it highlights the importance of future backscatter devices with high energy harvesting efficiency. The simulation results and discussion provided in this article will be helpful in understanding the coverage aspects of practical backscatter communication system in a smart city environment.

## 1. Introduction

The smart city is the integration of information and communication technology (ICT) into the classical city infrastructure in order to digitize the information and the processes that are involved in the management of a city [1]. In smart city, the data from different sources are collected by smart sensors, which often use the ecosystem of the Internet of Things (IoT) and, later, these data are analysed and shared with other systems. The collected data are not only used for efficient utilization of resources but also to reduce the operational cost of a city [1]. Therefore, the IoT has numerous applications in the monitoring of urban systems, such as water and electricity metering, controlling public street lights, updating public transport status, and waste management in public areas.

The IoT deployments for smart city applications only require the sensors to collect and transmit a small amount of data. In recent years, we have witnessed the development and deployment of low-power sensor networks of different radio technologies, such as LoRa, SigFox, NB-IoT, and Bluetooth low energy (BLE). These technologies have made a trade-off between the data rate and the coverage i.e., decreased the former to increase the latter, and the sensors utilizing these technologies are often powered with built-in batteries. However, in general, the large scale deployments of IoT networks of these technologies are mainly limited by cost, energy consumption, and the congestion of the communication medium [2]. In particular, their significant power consumption also brings forth the inconvenience of maintaining and replacing the batteries in order to accommodate a long range communication. Therefore, an alternative ICT that not only enables ultra-low-power communications rather also supports a large coverage area, while not overloading the scarce communication spectrum is needed, in order to realize the massive IoT deployments for smart cities.

Backscatter communication (BC) has recently emerged as an energy efficient ICT technology that provides optimized solutions to the requirements of smart city applications [3]. This radio technology alters the RF signal, impinging it at the antenna without using power-hungry and expensive radio frequency (RF) electronics so that significant cost and power savings are obtained. The BC devices can operate on licensed communication spectrum since they do not affect the operation of other systems [4]. These advantages come at the cost of significantly lower signal levels at the receivers and, thus, the data rate and the coverage of BC are limited [2]. Despite these limitations, the aforementioned advantages have led the researchers from both academia and industry to investigate the possibilities that are enabled by this ICT in application fields. such as smart agriculture, digital healthcare, and industrial automation [5]. To the best of our knowledge, the coverage of BC for large-scale deployment of BC for smart-city applications is not available in the literature and, thus, the feasibility of this ICT for smart city applications is not known.

### Targets and Key Contributions

In this article, we investigate the feasibility of BC, which is driven by a carrier signal that is generated by available cellular system base stations, in terms of coverage that can be achieved. In particular, we evaluate the outage ratio of a BC system based on the receiver sensitivity level of different low-power wide area technologies through a comprehensive set of simulations in an urban city environment. Here, the target is to evaluate the viability of cellular network based BC system in a downtown area of Helsinki city with practical system parameters. The simulation work is carried out by using a hybrid approach of deterministic ray tracing model, 3GPP indoor hotspot model, and ITU-R device-to-device (D2D) outdoor model.

The intent of this research work is to provide practical insights for a commercial *large-scale BC system* from the coverage planning and deployment point of view. Specifically, this article makes the following key contributions:The received signal power of the forward link i.e., between the transmitter (Tx) and the backscatter device (BD), in both indoor and outdoor condition is evaluated while utilizing an in-house built sophisticated 3D ray tracing simulator and a real city map data, at four different frequencies i.e., 200 MHz, 500 MHz, 700 MHz, and 900 MHz. The acquired simulation results shows that, for a considered case study, the mean received power levels at outdoor BDs are around –33 dBm, –42 dBm, –45.5 dBm, and –48 dBm at 200 MHz, 500 MHz, 700 MHz, and 900 MHz, respectively. Similarly, for indoor BDs, the received power level varies with respect to the frequency of operation and outdoor to indoor building penetration loss.The signal propagation at backscatter link i.e., between the BD and the receiver, is investigated in both the indoor and outdoor condition while utilizing 3GPP indoor hotspot model and ITU-R D2D outdoor model, respectively. For different combinations of forward and backscatter link distances, the received power at the reader is found at four considered frequencies. The obtained results for a considered case study configuration show that cellular based AmBC system is able to provide a significantly high received power to the reader in an outdoor environment at 200 MHz with BD-reader distance up to 60 m. Whereas, at 500 MHz, the mean received power at the reader is above –140 dBm for up to 50 m BD-reader distance in an outdoor environment. Promising coverage results are obtained for indoor BD communicating with the indoor reader at 200 MHz centre frequency. The received power level is clearly found higher than –140 dBm at indoor readers at 200 MHz frequency, even with high building penetration loss model.For energy harvesting, a significantly high received power level at the BD is required, therefore, the energy harvesting at the backscatter device is assessed at 200 MHz and 500 MHz frequency of operation in an outdoor environment, and only for LoRa backscatter technology. The obtained results show the time that is required for harvesting the energy as a function of different energy harvesting efficiencies.Finally, the coverage outage ratio with respect to the receiver sensitivity level of three radio technologies i.e., LoRa backscatter, NB-IoT, and BLE, is evaluated in both indoor and outdoor conditions. Interestingly, for a considered case study, a zero outage is found with LoRa and NB-IoT technology at 200 MHz in an outdoor environment for up to 60 m BD-reader distance. It is fascinating to learn from the obtained simulation results that, even at 900 MHz frequency, both LoRa and NB-IoT offer a zero outage ratio for BD-reader distance up to 35 m. It highlights the potential of LoRA backscatter and NB-IoT technology for those outdoorAmBC applications that do not require placing the reader more than 35 m away from the BD.

The remaining part of the article is organized, as follows. Section 2 outlines different low power wide area network technologies, and it briefly describes backscatter communication and its application in a smart city environment. Section 3 provides the details of three propagation models; those are considered for the simulation work of this article. Section 4 explains the simulation methodology adopted, and it gives details about the simulation environment, simulation cases, and simulation modeling parameters. Section 5 discusses the acquired simulation results. Finally, Section 6 concludes the article.

## 2. Background

### 2.1. Low Power Wide Area Network (LPWAN)

Low-power wide area networks (LPWANs) are characterised by low-power, long-range, and low-throughput wireless transmission technologies, mainly operating at sub-gigahertz (sub-GHz) frequency bands. When compared with the short-range transmission techniques, i.e., Wi-Fi, Bluetooth, NFC, ZigBee, etc., the LPWANs are capable of offering an adequate level of connectivity to the low-power devices that are distributed over a large geographical area. LPWAN can be considered to be a fundamental enabler of the IoT paradigm, and it is gaining momentum in the context of smart city and massive machine type communication (mMTC) in 5G and beyond. More specifically, LPWAN technologies are designed for a wide area coverage, and they are effective for both indoor and outdoor IoT or machine-to-machine (M2M) type scenarios. Ultra-low power operation enables the long-term stability, and makes it easy to maintain IoT devices for longer period of time, and it decreases the cost, energy consumption, and devices maintenance. Related experiments and their analysis on the coverage, throughput, and power consumption across different LPWAN technologies can be found at references [6,7,8,9,10].

#### 2.1.1. LoRa

This is one of the most popular and successful techniques, consisting of a protocol stack that is specified by LoRa Alliance [11]. LoRa operates over a long-range physical layer on unlicensed frequency bands while using the chirp spread spectrum (CSS) modulation technique, with the key features, such as low data rate, low complexity, and flexible operating classes for various applications, which enables a massive number of nodes per single gateway.

#### 2.1.2. SigFox

This was the first LPWAN technology that was proposed for an IoT market in 2009 [12]. It offers an end-to-end LPWAN connectivity solution by itself or by cooperating with other network operators. The physical layer of SigFox employs ultra-narrowband (UNB) modulation, and the end devices connect to the base station, which are equipped with cognitive software-defined radios (SDRs), utilizing binary phase shift keying (BPSK) modulation in an ultra-narrow sub-GHz ISM band [13]. Benefiting from UNB, SigFox is in possession of high spectrum efficiency and it experiences low noise levels and, thus, results in high receiver (Rx) sensitivity, ultra-low power consumption, and inexpensive antenna designs.

#### 2.1.3. NB-IoT

When compared with other LPWANs that use unlicensed band, NB-IoT utilizes the same frequencies as long term evolution (LTE), and it satisfies the need of IoT communication in a cellular system with low-complexity and low-throughput. To reduce device cost and minimise the battery consumption, LTE features, such as handover, carrier aggregation, and channel quality measurements, are removed in NB-IoT to simplify and optimize the communication procedure [14].

#### 2.1.4. Bluetooth Low Energy (BLE)

BLE is considered to be an important short-range radio technology with high market penetration i.e., it is already present in billions of existing smart devices, e.g., smart phones and tablets. BLE was initially introduced by Nokia in 2006 and, later, in 2010, Bluetooth Version 4.0 was added to the Core Specification [15]. Currently, Bluetooth Special Interest Group (SIG) provides standards for newer version of BLE [16]. There are two types of channels in the newer version of BLE i.e., advertising channels that are used for the purpose of broadcast and device discovery, and data channels that are used for bi-directional data transmission [16].

#### 2.1.5. Ingenu

Ingenu from the On-Ramp Wireless company [12], who developed and owned the random phase multiple access (RPMA) technique [17], is another emerging LPWAN technology. Different from the most of LPWANs using sub-GHz, Ingenu works at 2.4 GHz band. The robust physical layer design of Ingenu enables long-range radio link under challenging RF environments.

#### 2.1.6. DASH7 Alliance Protocol (D7AP)

This is an open-source wireless sensor and actuator network (WSAN) protocol with a full seven-layer OSI stack from physical layer to application layer [18]. D7AP provides a long-range coverage i.e., up to 2 km, with low-latency transmission and long battery life. Burst transmission, light packet, asynchronous communication, and good mobility are the main features of D7AP [6].

### 2.2. Backscatter Communication

#### 2.2.1. Traditional

In a traditional BC, a transmitter indirectly communicates with the receiver, which is also referred to as a reader along with a passive or semi-passive backscatter device in the middle. In case of BC, a dedicated transmitter generates a radio signal, which is reflected/modulated by the BD and is then received at the reader. The received signal at the BD from the transmitter can be used for two purposes i.e., to power up the circuitry of the backscatter device, and to carry the information to the reader in the form of the modulated signal. There are three main modes of traditional backscatter communication i.e., monostatic, bi-static co-located, and bi-static dislocated configuration, as shown in Figure 1. Each BC mode possesses different link budget and margins, as explained in references [19,20,21]. First, in monostatic backscatter mode, a single antenna is used at the reader for both transmission and reception purpose, where the forward link is from the transmitter to BD and the backscatter link is from BD to the Rx as shown in Figure 1a. The main challenge of monostatic architecture is self-interference, and that limits the range of data transmission. To improve this, active cancellation can be implemented to remove the self-interference signal. Second, in a bi-static co-located mode, the reader has two antennas; one for transmission and one for reception, and they are located quite close to each other i.e., separated by few wavelengths (lambdas), as shown in Figure 1b. In bi-static co-located mode, the Tx antenna and Rx antenna of the reader may have different gains. Third, in bi-static dislocated mode, the Tx and the Rx are placed at two separate locations, as demonstrated in Figure 1c. There also exists a direct path between the Tx and the Rx, different from the backscatter path, and the direct path is much stronger than backscatter path [19,20,21].

#### 2.2.2. Ambient

Ambient backscatter communication (AmBC) utilizes the ambient RF signal to realise a communication between BD and the reader, and the source of signal at BD can be any existing RF signal, as demonstrated in Figure 1d. The main advantage of the AmBC is that it utilizes the abundant existing radio frequency signal (e.g., cellular network, WLAN, TV, and radio broadcast) for transmitting information from the BD to the Rx, and also for harvesting the energy at the BD. More specifically, ambient BD needs no dedicated energy provider, and it reduces the cost of installation and maintenance. Also, ambient backscatter enables D2D communication, different from the traditional backscatter that must communicate exclusively with a reader, without sensing and communicating with other nearby BDs. Vincent et al. [22] discussed a fundamental scenario of AmBC, where two battery-free devices communicate via ambient backscatter.

### 2.3. Application of Backscatter Communications

Many applications based on traditional BC were studied, such as tracking devices, remote switches, medical telemetry, and sensor networks. Recently, with the studies and development on AmBC, many applications are emerging in various industries, such as smart life, medical and healthcare, retail and logistics, and transportation:

(i) Smart Home: ambient BDs can be deployed in a home to improve a life quality. A large amount of backscatter sensors can be placed at home [23], which can operate for a long time without dedicated power sources. These backscatter sensors can detect normal environmental indices (e.g., temperature, humidity) and harmful gases (e.g., CO, smoke) to prevent potential risks at home.

(ii) Medical and Healthcare: tiny-scale and long-lasting smart devices are required for medical and healthcare applications, such as wearable and implantable health monitoring, where ambient BDs can be implemented [5]. Huang et al. [24] developed a battery free platform for wearable devices such as smart sneaker, which embedded sensors and ambient backscatter modules in the shoes. Sensors can count steps and heart rate, and two shoes are coordinated by using the ambient backscatter modules. Wang et al. [25] proposed a smart fabric, where a BD inside a shirt monitors vital signs of human body, such as heart rate and breathing rate.

(iii) Retail and Logistics: ambient BDs can be deployed in retail and logistics. In the study of Vincent et al. [22], each item in a grocery store is equipped with a BD that has a unique identification number (ID). One BD broadcasts its ID in every five-second interval, and all BDs in the same network periodically listen and store their neighbour BDs. By this means, the existence of an item can be indicated by comparing its ID with the information stored in the neighbours. Furthermore, this technique has been implemented in self-service retail. For example, each item in a vending machine is stuck with a BD label. Buyers are required to register a payment method (e.g., credit card, PayPal) before purchase. The payment is automatically executed when the item is taken out of the vending machine. The similar solution can also be implemented in logistics, e.g., warehouse and container.

(iv) Transportation: backscatter communication enables reliable, accurate, and low-cost vehicular positioning in high-density cites, tunnels, and underground scenarios, where the global navigation satellite system (GNSS) has an unacceptable performance due to signal shielding. Han et al. [26] proposed a novel BC assisted vehicular positioning system, where a backscatter reader is deployed on a vehicle and backscatter tags with location information are deployed along roadside. Thus, the vehicle can be located by BC between the reader and tags. This technique can achieve high positioning accuracy and it can be easily expanded. In the high-speed railway (HSR) industry, Dou et al. [27] developed a backscatter aided wireless transmission scheme.

### 2.4. Energy Harvesting

Backscatter devices usually work in a passive or semi-passive mode. There exists an internal battery in the semi-passive BD for powering up the circuitry, and that provides a longer lifetime and communication range to the backscatter device. In addition, the larger form-factor of the BDs considerably contributes to their applications in different industries. Whereas, in the passive mode, the BD utilizes the power of the incident RF signal to harvest the energy while there is no energy reservoir included at the BD. The battery-free BD requires the energy to process the obtained information and transmit the information to the reader. The form-factor of these kind of BDs is tiny. Passive BDs can obtain sustainable power through ambient signals that are readily available in abundance in the environment. The ubiquitous availability of ambient RF signals makes it feasible to harvest the RF power. The amount of energy available for harvesting at the BD depends on the received signal power level from the RF source. There is a trade-off between the harvested energy and decode information rate. Different schemes, e.g., time splitting, static power splitting, and on-off splitting, have been studied for allocating the RF power for harvesting the energy in different scenarios, and it is recommended to use static power splitting in high signal-to-noise (SNR) regions and on-off splitting scheme in intermediate SNR regions [28].

### 2.5. Related Work

Although significant research progress and developments have been been made in the domain of backscatter communications, to the best of our knowledge, there is currently no study available on the coverage aspects of a BC system based on a cellular generated carrier. The reference [13] provides an overview of LPWAN, whereas Vejlgaard et al. provides the coverage and capacity analysis of several LPWAN technologies in an indoor and outdoor environment using existing cellular sites [29]; however, a backscatter communication system was not considered in their analysis. The topic of BC and AmBC is catching the attention of the researchers in both industry and academia, and the number of studies and surveys related to AmBC are now available in the literature [2,22,30,31]. The concept of harvesting the energy from RF signal is also well supported in case of backscatter communication, and the research on the energy harvesting can be witnessed in references [32,33]. The trade-off between the data rate and the quantity of harvested energy were displayed and discussed in [28]. A BC scenario for cognitive wireless networks, as developed by Hoang et al. [34], proposed an effective method to improve the network performance that a secondary user powered by radio signals take advantage of the transmission energy of the primary user. A game-theoretic time allocation problem was proposed to improve the network throughput, and an optimal time allocation ratio of energy harvest and backscatter was obtained. Kim et al. [35] conducted research on multiple access (MA) hybrid backscatter communication to transmit radio power.

The work by Lu et al. [36] discussed the bit-error-rate (BER) when detecting one-bit information in the backscatter communication network. The energy requirements of each channel to detect this information were analysed. Hoang et al. [31] proposed a fixed channel model for information transfer and energy harvest in a cognitive wireless network. The active information transmission of the BD is optimized for an optimal throughput in their work. Users are capable of collaborating by using radio power transmission, according to Xun et al. [37]. They displayed performance improvement of the network results from backscatter-assisted cooperation, compared with the normal active information transmission. Choi et al. developed a MAC protocol that operates with network devices and BDs [38] to improve the network throughput and energy efficiency. Wu et al. [39] investigated the beamforming design optimization problem that maximizes the transmission capacity, and provided an exact penalty method that can achieve the capacity closed to the upper bound. Tao et al. [40] discussed the interference cancellation problem of Bistatic backscatter communication system and developed a optimal detector to cancel the direct path interference. Felisberto et al. established good practices for the design of IoT devices with a focus on two main design challenges i.e., power consumption and the connectivity, and discussed the challenges of resource constrained IoT devices [10]. In this work, we have used the sensitivity levels and the receiver design criteria of different existing technologies, i.e., LoRa backscatter, NB-IoT, and BLE5, for evaluating the coverage and feasibility of a cellular generated carrier based backscatter communication system in our case study.

## 3. Propagation Models

### 3.1. Ray Tracing

Accurate radio channel characterization and coverage prediction methodology are required in order to efficiently plan a communication network. Ray tracing (RT) is a deterministic approach for radio propagation modeling, and it is widely used for detailed channel characterisation and for coverage prediction in various types of environments [41,42]. Besides accurate modeling, RT offers a notable advantage: it is less dependent on the bandwidth and carrier frequency than stochastic models that are based on measurements. Unlike other empirical and semi-empirical channel models that rely on averaged channel measurements, the deterministic approach of RT utilizes geometrical optics (GO) and case-specific three-dimensional (3D) propagation environment [41]. RT also requires the information about the physical properties of the objects that are present in the environment, but, on the other hand, available ray tracing models are credible and accurate. Recent surveys on RT can be found from [43,44,45,46]. The computational load and complexity of the RT model are higher when compared with other empirical and semi-empirical models. Furthermore, the computational load of the RT approach increases with the complexity of the simulation environment [47]. However, nowadays, RT methods have gained a lot of interest due to the availability of servers and clusters with high computational capabilities. In the literature, two broad classes of RT algorithms can be found based on their implementation i.e., image theory (IT) method [47] and shoot and bouncing ray (SBR) method [48], also known as ray launching (RL). The IT method is suited for a less complex environment, whereas the SBR method is more robust and better suited for complex environment [49]. Algorithm 1 shows the pseudocode of the ray tracing model.
**Algorithm 1** 3D ray tracing model
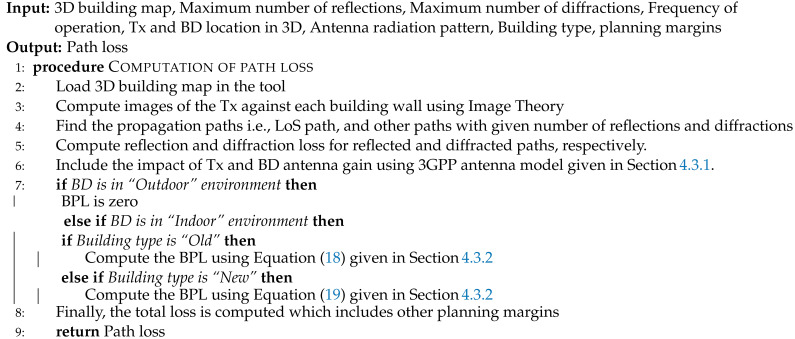


### 3.2. ITU Outdoor D2D Model

The radio communication sector of international telecommunication union (ITU-R) provides the guidance and recommendations for outdoor short range communication. In a technical report ITU-R P.1411-10 [50], the ITU-R presents a path loss model for outdoor D2D type of communication, where the devices can be located below the roof-top height close to the street level. The ITU-R presents a general model for both line of sight (LoS) and non-LoS (NLoS) conditions. The path loss in LoS is given by Equation (Equation 1), where $PL_{LoS,dB}$ is the path loss in dB value for LoS condition, $PL_{LoS,dB}^{Median}$ is the median value of path loss in LoS case, and $\Delta_{LoS}$ is the LoS location correction factor. $PL_{LoS,dB}^{Median}$ is calculated, as given in Equation (Equation 2), where $f_{c,MHz}$ is the carrier frequency in MHz, and $d_{km}$ is the separation between the terminals/devices in kilometers. The $\Delta_{LoS}$ is the function of the location percentage *p*, and it includes the impact of shadowing, as shown in Equation (Equation 3). Shadowing is generally modelled as a Gaussian random variable with zero mean and $\sigma_{LoS}$ is the standard deviation of the shadowing in LoS condition. The reported value of $\sigma_{LoS}$ is 8.5 dB at [50]. ITU defines location percentage as the percentage of locations in which the the predicted path loss exceeds the actual path loss. The location percentage can have any value between 1–99%.
(1)$$PL_{LoS,dB}=PL_{LoS,dB}^{Median}+\Delta_{LoS}$$
(2)$$PL_{LoS,dB}^{Median}=32.45+20\log_{10}\left(f_{c,MHz}\right)+20\log_{10}\left(d_{km}\right)$$
(3)$$\Delta_{LoS}=1.5624\sigma_{LoS}(\sqrt{-2\ln\left(1-p/100\right)-1.1774)}$$

Similarly, the path loss in NLoS condition can be computed using Equation (Equation 4), where $PL_{NLoS,dB}$ is the path loss in dB value for NLoS condition, $PL_{NLoS,dB}^{Median}$ is the median value of path loss, and $\Delta_{NLoS}$ is the location correction factor for NLoS condition, respectively. $PL_{NLoS,dB}^{Median}$ is calculated, as given in Equation (Equation 5), where $L_{urban}$ is the environment correction factor, and the recommended value of $L_{urban}$ is 0 dB and 6.8 dB for suburban and urban environment, respectively [50]. For a given location percentage, the $\Delta_{NLoS}$ can be computed by Equation (Equation 6), where $\sigma_{NLoS}$ is the standard deviation of shadowing in the NLoS condition and $N^{-1}\left(.\right)$ is the inverse normal cumulative distribution function. The recommended value of $\sigma_{NLoS}$ is 8.5 dB given at reference [50].
(4)$$PL_{NLoS,dB}=PL_{NLoS,dB}^{Median}+\Delta_{NLoS}$$
(5)$$PL_{NLoS,dB}^{Median}=9.5+45\log_{10}\left(f_{c,MHz}\right)+40\log_{10}\left(d_{km}+L_{urban}\right)$$
(6)$$\Delta_{NLoS}=\sigma_{NLoS}N^{-1}\left(p/100\right)$$

Finally, the basic path loss at any distance *d* with location percentage *p* is given as:(7)$$PL\left(d,p\right)=\left\{\begin{array}{c} PL_{LoS,dB}\left(d,p\right)\;\;\;\;\;\;\;\;\;\;\;\;\;\;\;\;\;\;\;\;\;\;\;\;\;\;\;\;\;d<d_{LoS}\\ PL_{NLoS,dB}\left(d,p\right)\;\;\;\;\;\;\;\;\;\;\;\;\;\;\;\;\;\;\;\;\;\;\;\;\;\;\;d>d_{LoS}+w\\ PL_{T,dB}\left(d,p\right)\;\;\;\;\;\;\;\;\;\;\;\;\;\;\;\;\;\;\;\;\;\;\;\;\;\;\;\;\;\;\;\;\;else \end{array}\right.$$
where, in Equation (Equation 7), $d_{LoS}$ is the LoS distance for which the $PL_{LoS,dB}\left(d,p\right)$ is applicable, and it is the function of location percentage *p*, as shown in Equation (Equation 8), *w* is the transition distance for the device to move away from LoS condition to completely NLoS condition, and $PL_{T,dB}\left(d,p\right)$ is the basic path loss for transition state distance i.e., for $d\geq d_{LoS}$ and $d\leq d_{LoS}+w$, where $PL_{T,dB}\left(d,p\right)$ is computed by linear interpolation of path loss between the LoS and NLoS condition, as shown in Equation (Equation 9).
(8)$$d_{LoS}\left(p\right)=\left\{\begin{array}{c} 212{\left[\log_{10}\left(p/100\right)\right]}^{2}-64\log_{10}\left(p/100\right)\;\;\;\;\;\;\;\;\;\;\;\;\;\;\;\;\;\;\;\;\;\;\;\;\;\;\;\;\;p<45\\ 79.2-70(p/100)\;\;\;\;\;\;\;\;\;\;\;\;\;\;\;\;\;\;\;\;\;\;\;\;\;\;\;\;\;\;\;\;\;\;\;\;\;\;\;\;\;\;\;\;\;\;\;\;\;\;\;\;\;\;\;\;\;\;\;\;else \end{array}\right.$$
(9)$$PL_{T,dB}\left(d,p\right)=PL_{LoS,dB}\left(d_{LoS},p\right)+\left(PL_{NLoS,dB}\left(d+w,p\right)-PL_{LoS,dB}\left(d,p\right)\left(d-d_{LoS}\right)/w\right)$$

Figure 2 shows the basic path loss for an urban environment at 500 MHz frequency as a function of distance for different values of location percentage (p). The transition distance (w) of 20 m was assumed in Figure 2. Algorithm 2 presents the pseudocode of the ITU outdoor D2D propagation model.
**Algorithm 2** ITU outdoor D2D propagation model
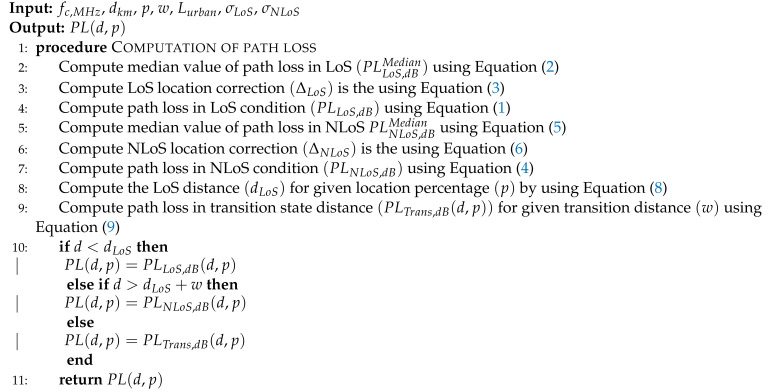


### 3.3. 3GPP Indoor Hotspot (InH) Model

The 3GPP channel modeling community carried out a study on channel models for frequencies up to 100 GHz, and it provides a path loss model for indoor environment in technical report TR 38.901 [51]. The report provides an indoor path loss model for two different types of conditions i.e., LoS and non-LoS (NLoS) condition, as given in Equations (Equation 10) and (Equation 11), respectively. The 3GPP InH model consists of a distance and frequency dependent deterministic part and a random part that includes the impact of shadowing. In Equation (Equation 10), $PL_{LoS,dB}$ is the path loss in dB for LoS condition, $d_{3D,m}$ is the three dimensional distance between the two points in meters, $f_{c,GHz}$ is the frequency of operation in GHz, and $\zeta_{LoS,dB}$ is the shadowing/slow fading factor in dB under the LoS condition, and the shadowing factor in LoS is modelled as a Gaussian random variable with zero mean and standard deviation $\sigma_{LoS}$, where $\sigma_{LoS}=3$ for $1m\geq d_{3D,m}\leq 150m$. Moreover, in Equation (Equation 11), $PL_{NLoS,dB}$ and $\zeta_{NLoS,dB}$ are the path loss and shadowing factor in dB for NLoS condition, respectively, and $\zeta_{NLoS,dB}$ is modelled as a Gaussian random variable with zero mean and standard deviation $\sigma_{NLoS}$, where $\sigma_{NLoS}=8.03$ for $1m\geq d_{3D,m}\leq 150m$.
(10)$$PL_{LoS,dB}=32.4+17.3\log_{10}\left(d_{3D,m}\right)+20\log_{10}\left(f_{c,GHz}\right)+\zeta_{LoS,dB}$$
(11)$$PL_{NLoS,dB}=\max\left(PL_{LoS,dB},17.3+38.3\log_{10}\left(d_{3D,m}\right)+24.9\log_{10}\left(f_{c,GHz}\right)+\zeta_{NLoS,dB}\right)$$

The 3GPP report defines two types of indoor environments i.e., open and mixed indoor types, and the LoS probability is defined for each indoor environment types, as follows:

#### 3.3.1. Open Indoor

(12)$$Pr_{LoS}=\left\{\begin{array}{c} 1\;\;\;\;\;\;\;\;\;\;\;\;\;\;\;\;\;\;\;\;\;\;\;\;\;\;\;\;\;d_{2D}\leq 5m\\ e^{-\frac{d_{2D}-5}{70.8}}\;\;\;\;\;\;\;\;5m\geq d_{2D}\leq 49m\\ 0.54e^{-\frac{d_{2D}-49}{211.7}}\;\;\;\;\;\;\;\;\;\;d_{2D}>49m \end{array}\right.$$

#### 3.3.2. Mixed Indoor

(13)$$Pr_{LoS}=\left\{\begin{array}{c} 1\;\;\;\;\;\;\;\;\;\;\;\;\;\;\;\;\;\;\;\;\;\;\;\;\;\;\;\;\;\;\;\;\;\;d_{2D}\leq 1.2m\\ e^{-\frac{d_{2D}-1.2}{4.7}}\;\;\;\;\;\;\;\;1.2m\geq d_{2D}\leq 6.5m\\ 0.32e^{-\frac{d_{2D}-6.5}{32.6}}\;\;\;\;\;\;\;\;\;\;\;\;\;\;d_{2D}>6.5m \end{array}\right.$$

Thus, according to [52], at any given distance, the path loss along with shadowing in an indoor environment $PL_{InH,dB}$ under 3GPP InH model is given by Equation (Equation 14)
(14)$$PL_{InH,dB}=Pr_{LoS}\cdot PL_{LoS,dB}+\left(1-Pr_{LoS}\right)\cdot PL_{NLoS,dB}$$

Figure 3 shows the path loss that is acquired through 3GPP InH model at 500 MHz in an open and mixed type indoor environment, and the path loss that is acquired through free space path loss (FSPL) model is also shown in Figure 3 for reference. The pseudocode of the ITU outdoor D2D propagation model is given as Algorithm 3.
**Algorithm 3** 3GPP Indoor Hotspot (InH) propagation model
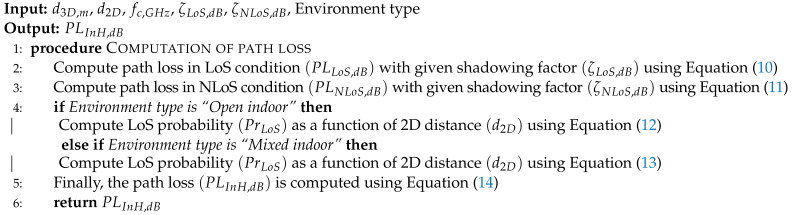


## 4. System Design and Simulation Configuration

### 4.1. Environment and System Description

We consider a practical AmBC system in which the cellular network acts as a source of ambient signal, and several BDs and readers are located in an outdoor and indoor urban city environment. While the cellular network communicates with its actual users e.g., smartphones, laptops, TV, etc., both BD and the reader may also receive the signal from the ambient source. For simulations, we have considered a three-dimensional building data of the downtown area from Helsinki city. The area under consideration has buildings and structures with irregular shapes and heights, and roads and streets with varying width. The buildings are of multiple stories ranging from two to eight floors. The target area is covered with ten three-sectored macro sites, where the location of macro sites were selected with careful planning and consideration. The transmit antenna of each sector is mounted at a height of 30 m above the ground, and it is above the average height of buildings’ rooftop. Figure 4 shows the two dimensional map of the area under consideration, and it highlights the location of each site and the azimuth direction of each sector.

Our target is to investigate the performance of bi-static dislocated mode of BC in a real world urban city environment i.e., Helsinki, a capital city of Finland. For this purpose, 750 static BDs with 1.5 m height are homogeneously and randomly distributed in a target area, as shown in Figure 5. Out of those 750 static backscatter devices, 113 BDs are placed in an outdoor environment, as marked with blue circles, and 667 BDs are positioned in an indoor environment at different floors as marked with orange circles in Figure 5. As mentioned earlier, unlike regular Manhattan type building grid with equal building heights and street spacing, we have considered a real world city environment with variable building heights, and the indoor BDs are distributed at four different floors i.e., ground floor (1.5 m), second floor (7.5 m), fifth floor (16.5 m), and seventh floor (22.5 m), depending upon the height of the building. Table 1 presents the distribution of the BDs at different floors.

### 4.2. Simulation Cases

We have considered mainly two cases here:

Outdoor BD with outdoor Rx: in this case, the BD and the Rx are both located in an outdoor environment. The ray tracing model and ITU outdoor D2D model are used to find the propagation path loss of forward link and backscatter link, respectively. Indoor BD with indoor Rx: in this case, the BD and the Rx are both located in an indoor environment. A RT model is used as a propagation model for the forward link and 3GPP InH model is used for backscatter link, respectively. We have considered a mixed type indoor environment, and computed the received power at the indoor receiver assuming old and new building type.

### 4.3. Simulation Tool, Models and Parameters

All of the propagation and other models are implemented in MATLAB platform for simulations. First, a full three-dimensional (3D) ray tracing tool based on image theory was developed by the authors for finding the propagation paths between the transmitter and the BD. In our simulations, we utilize directional antennas at the Tx i.e., cellular antenna; therefore, the information about the directions of the propagation path is required for the forward link. The RT tool not only provides the path loss, but it also gives the delay and angular information of the propagation paths. We have considered the propagation paths with two reflections and one diffraction at maximum. Similarly, MATLAB is used for implementing ITU outdoor D2D model and 3GPP InH model for computing the path loss in the backscatter link. Details of the other models considered in the simulation are given in forthcoming subsections.

#### 4.3.1. 3GPP Antenna Model

In this study, macro cellular communication antennas are considered to be a source of the ambient signal. In a considered layout of the cellular network shown in Figure 4, each cell is equipped with directional antenna, and we have used the extended 3GPP antenna model that is presented in [51] to model the antenna radiation pattern in azimuth and elevation plane. The antenna gain G(φ,θ) for any horizontal direction φ and vertical direction θ is given by Equation (Equation 15), where $G_{H}\left(\varphi \right)$ and $G_{V}\left(\theta \right)$ are gains of the antenna in the azimuth and elevation plane in dB scale, respectively. In Equation (Equation 16), $-180^{\circ }\leq \varphi \leq 180^{\circ }$, $HPBW_{H}$ is the half power beamwidth (HPBW) of the antenna in degrees in the horizontal plane, $FBR_{H}$ is the front to back ratio (FBR) in dB scale, and $\varphi_{az}$ is the reference azimuth direction of the antenna in degrees, and $A_{Max}$ is the maximum gain of the antenna [dBi]. Whereas, in Equation (Equation 17), $-90^{\circ }\leq \theta \leq 90^{\circ }$, $HPBW_{V}$ is the HPBW of the antenna in degrees in the vertical plane, $SLL_{V}$ is the side lobe level (SLL) of the radiation pattern in vertical domain in dBs, and $\theta_{tilt}$ is the reference downtilt angle of the main beam of the antenna. Table 2 provides the antenna radiation modeling parameters used in our simulations.
(15)$$G\left(\varphi ,\theta \right)=G_{H}\left(\varphi \right)+G_{V}\left(\theta \right)$$
(16)$$G_{H}\left(\varphi \right)=-\min\left[12\left(\frac{\varphi -\varphi_{az}}{HPBW_{H}}\right),FBR_{H}\right]+A_{Max}$$
(17)$$G_{V}\left(\theta \right)=-\min\left[12\left(\frac{\theta -\theta_{tilt}}{HPBW_{V}}\right),SLL_{V}\right]$$

#### 4.3.2. Building Penetration Loss (BPL) Model for Indoor BD

In our simulations, the source of an ambient signal is located in an outdoor environment. Therefore, for indoor BDs we need to consider the building penetration loss (BPL). The 3GPP has defined the penetration loss for different types of materials as a function of the frequency for two types of buildings i.e., for old and new building type, such that old and new buildings have lower and higher BPL, respectively [51]. Here we have assumed that old buildings are composed of 15% plain glass windows and 85% concrete wall, whereas the modern buildings are covered with 70% infrared reflective (IRR) glass windows and 30% concrete wall. Accordingly, we are using a modified form of those BPL models presented by 3GPP, see Equations (Equation 18) and (Equation 19). In Equation (Equation 18), $BPL_{Low,dB}$ is the BPL for the old building type and is expressed in dB, $L_{Plainglass,dB}$, and $L_{Concrete,dB}$ are the single material penetration losses for plain glass and concrete in dB, respectively. Similarly, in Equation (Equation 19), $BPL_{High,dB}$ is the BPL for the modern building type and $L_{IRRglass,dB}$ is the penetration loss for IRR glass in dB. The penetration loss for double plain glass, IRR glass and concrete slab are given in Equations (Equation 20)–(Equation 22), respectively, where *f* is the frequency that is expressed in GHz [51].
(18)$$BPL_{Low,dB}=-10\log_{10}\left[0.15\times 10^{\frac{-L_{Plainglass,dB}}{10}}+0.85\times 10^{\frac{-L_{Concrete,dB}}{10}}\right]$$
(19)$$BPL_{High,dB}=-10\log_{10}\left[0.7\times 10^{\frac{-L_{IRRglass,dB}}{10}}+0.3\times 10^{\frac{-L_{Concrete,dB}}{10}}\right]$$
(20)$$L_{Plainglass,dB}=2+0.2f$$
(21)$$L_{IRRglass,dB}=23+0.3f$$
(22)$$L_{Concrete,dB}=5+4f$$

### 4.4. General System Parameters

We recall that the aim is to investigate the feasibility of AmBC in a real world urban environment. Therefore, we model the simulation environment using realistic parameters and assumptions that reflect a practical AmBC system in order to acquire rational and practical results. The performance metrics considered for the analysis are the received power at the BD, the received power at the Rx, the time required for harvesting the energy at BD, and the outage ratio. Cellular antennas act as a source of the ambient signal, while the propagation between BD and Rx defines the coverage and outage of the service. The parameters applied in the simulations are listed in Table 3. Table 2 provides the parameters of the transmit antenna radiation pattern model, where a 0 dBi antenna gain is assumed for the BD in the forward and backscatter link, and also for the Rx antenna. The target of this study is to cover the existing cellular frequencies and the candidate frequency bands for IoT networks. Here, it must be noted that 900 MHz band is mainly used by cellular systems, and a large number of existing cellular sites use this band for providing basic cellular coverage. In Europe, the 700 MHz spectrum is available for digital television broadcasting [53], and is also considered to be a potential band for deploying 5G of cellular system [54]. Currently, in some European countries there is a debate going on about the usage of 200 MHz and different proposals are given, and the use of IoT at 200 MHz is one of them. The 500 MHz is considered to be an intermediate band between 200 and 700 MHz band. Therefore, for the research work of article, the simulations are performed at four different frequencies i.e., at 200 MHz, 500 MHz, 700 MHz, and 900 MHz. For indoor BDs, the results are presented with low and high BPL models and a in mixed type indoor environment. A fixed transmit power of 43 dBm is assumed for all macro cells in the network. We follow the reference [19], where, for bi-static dislocated type of backscatter communication with 5% outage probability and Rician factor K= 3 dB, the recommended fast fading margins are 10 dB and 5 dB for the forward and backscatter link, respectively. Similarly, for the forward link, a 7 dB slow fading margin is considered, and for the backward link the slow fading margins are selected following the recommendations given for the ITU D2D outdoor PL model and 3GPP InH model in [50,51], respectively. It is assumed that the polarization of the RF signal is not known; therefore, a polarization mismatch loss of 3 dB is considered in the forward link as well as in the backscatter link separately. A maximum power transfer is assumed, i.e., transmission co-efficient at BD is 1, and a 6 dB modulation loss factor at BD is also included in our computation. The receiver sensitivities of long range (LoRa) backscatter [55], narrow band-IoT (NB-IoT) [14], and Bluetooth low energy 5 (BLE 5) [16] technology are defined as –149, –141, and –97 dBm, respectively, as shown in Table 4. Backscatter devices can operate in a passive mode or in a semi-passive mode. It is reported in the reference [55] that a LoRa backscatter device requires 9.25 μW power for operation, and it can harvest the required energy from the dedicated or ambient source. The power consumption for NB-IoT and BLE 5 devices are 543 mW and 20 mW, respectively, and they typically admit a battery source for powering up the BD. In the case of BLE 4.0 compatible BLE-backscatter devices, the power consumption is improved to 1.55 mW, whereas the maximum data rate is limited to 1 Mbps with −92 dBm sensitivity level [56]. However, here, it must be noted that we have only focused on BLE 5 in this article. The sensitivity level, power consumption, and the maximum supported data rates of different IoT technologies that are considered in this article are given in Table 4.

## 5. Results and Discussion

This section discusses the acquired simulation results that are based on the simulation setup, network layout, and parameters presented in Section 4. The results for outdoor and indoor cases are presented separately. The first performance metric considered is the forward link received power at the backscatter device. Figure 6a,b show the cumulative distributive functions (CDFs) of the received power at four different frequencies for the outdoor and indoor BDs. In the case of indoor results, the solid and dashed lines represent the cases with the low and high building penetration loss model, respectively. Figure 6a shows that, in the considered environment and a cellular network layout, the received power at the outdoor BD is significantly high, even at 900 MHz. In an outdoor environment, the mean received power levels are around –33 dBm, –42 dBm, –45.5 dBm, and –48 dBm at 200 MHz, 500 MHz, 700 MHz, and 900 MHz, respectively. It shows that there is almost 9 dB difference in the received power at the BD between the 200 MHz and 500 MHz frequency of operation, and the difference in received power becomes smaller while migrating from 500 MHz to 700 MHz. Whereas, the indoor BDs received a clearly lower power when compared with outdoor BDs due to additional building penetration loss for indoor BDs. Moreover, it can be seen from Figure 6b that the high BPL model offers around 6–6.5 dB higher loss value as compared with the low BPL model, and the mean received powers at the BD in indoor environment using high BPL are around –44.5 dBm, –54.5 dBm, –58.5 dBm, and –61 dBm at 200 MHz, 500 MHz, 700 MHz, and 900 MHz, respectively.

In this article, we have only considered a LoRa backscatter device for energy harvesting. Furthermore, it is shown in the reference [57] that there are numerous factors that affect the energy harvesting efficiency of the BD. Therein, the energy harvesting efficiency was also shown as a function of the incident RF power. Accordingly, we have computed the required time for harvesting the energy with different energy harvesting efficiencies, starting from 40% to 80%.

Figure 7 shows the CDFs for the time that is required for harvesting the energy from the incoming RF signal from the transmitter at the outdoor LoRa backscatter device, while considering different energy harvest efficiencies and frequencies. Figure 7a shows that, for the considered simulation scenario at 200 MHz frequency of operation, the mean time that is required for harvesting the energy with 40% and 80% energy harvesting efficiency is around 86 s and 43 s, respectively, which seems viable and practical for BDs that need to send periodically small amount of data. However, in Figure 7b, the time that is required for LoRa BD at 500 MHz with 40% and 80% energy harvesting efficiency is around 880 s and 440 s, respectively, which is considerably high. For reducing the harvesting time, outdoor BDs can also be implemented with components that can harvest the energy from sources other than incoming RF signal, e.g., solar and wind energy. whereas, indoor BDs can take the advantage of the energy available from indoor lights fluorescent and indoor RF signals, such as TV and WiFi. Currently available BDs require quite high received power level e.g., –25 dBm for harvesting the energy, and the results that are presented here emphasize the need of developing the BDs that can harvest the energy with good efficiency with considerably low received power levels for supporting the battery free deployment of BDs.

The received power at the reader is the next metric that is considered here, as it defines the coverage and outage of the AmBC system. Figure 8a shows the mean received power at the reader for the case of outdoor BD with outdoor reader, where the *x*-axis is the distance between the BD and the reader in meter, and *y*-axis is the received power at the BD in dBm. It can be seen that, by increasing the distance between the BD and the reader, the mean received power decreases. The slope of the graph changes due to the LoS probability, as in the case of ITU outdoor D2D model the LoS probability is defined as a function of distance between the BD and the reader, and a transition distance of 20 m is used in our simulations. From the acquired results, it is clearly evident that the cellular based AmBC system is able to provide a significantly high received power to the reader in an outdoor environment at 200 MHz with BD-reader distances up to 60 m. However, at 500 MHz, the mean received power is above –140 dBm for the outdoor BD-reader up to 50 m distance. For the medium range distance, i.e., up to 40 m between the BD and a reader, a BC link with at least –140 dBm receiver sensitivity can be obtained by using, as an ambient source, an existing cellular network operating at 900 MHz.

Similarly, Figure 8b shows the mean received power at the reader while using low and high BPL in the case of indoor BD communicating with the indoor reader. For up to 60 m BD-reader separation the mean received power level is clearly higher than –140 dBm for indoor readers at 200 MHz centre frequency, even with high BPL. It highlights the usage of 200 MHz band for IoT and machine type communication in an urban environment for long and medium range distances. The radio propagation condition is slightly challenging at 500 MHz for indoor BD and reader as the mean received power is almost –140 dBm at 40 m BD-reader separation with high BPL model, and the coverage area shrinks with higher frequencies. Similarly, for other frequencies and BPL models, the mean received power levels at the reader can be found in Figure 8b.

The outage ratio is defined as the ratio of the locations with a received power below a receiver sensitivity threshold to the total number of receiver points considered in the area. Here, we have considered three thresholds i.e., –149 dBm, –141 dBm, and –97 dBm with respect to the receiver sensitivity of LoRa, NB-IoT, and BLE 5 technology, respectively, for computing the outage ratio of the considered AmBC system. Figure 9a–f show the outage ratio for outdoor and indoor BDs, respectively, while considering different backscatter technologies. Interestingly, it was found that there is zero outage with LoRa and NB-IoT at 200 MHz in outdoor environment, even for 60 m BD-reader distance, and that shows that both LoRa and NB-IoT are good candidate technologies for deploying AmBC network at 200 MHz in an urban macro cellular outdoor environment. At other higher considered frequencies, the outage ratio increases with the increase in BD-reader distance. However, it is fascinating to see that even at 900 MHz frequency, both LoRa and NB-IoT offers zero outage ratio for BD-reader distance until 35 m, it means that LoRA backscatter and NB-IoT can be effectively used for those outdoor AmBC applications that do not require placing the reader much further away e.g., less than 35 m from the BD. However, Figure 9c shows that in an outdoor environment the BLE 5 is only effective for very short distance between the BD and the reader i.e., up to 10 m at 200 MHz, otherwise the outage ratio is found to be quite high for BLE 5 at higher frequencies. Whereas, for indoor BDs, the BLE 5 is found to be highly ineffective due to a significantly higher outage ratio, as shown in Figure 9f. LoRa and NB-IoT show impressive outage results, even for indoor BDs operating at 200 MHz frequency, and a zero outage is ensured with LoRa for BD-reader distance of up to 60 m, whereas the NB-IoT shows the outage ratio of only 1.4% at 60 m BD-reader distance. LoRa still shows promising outage results at 500 MHz in a low-loss building type. Clearly, for NB-IoT, the 700 MHz and 900 MHz bands are not suitable for the cellular AmBC system with indoor BD and reader; however, LoRa backscatter can be used for short range up to 20 m BD-reader distance with a low outage ratio.

## 6. Conclusions

With the rapid worldwide urbanization, the need to efficiently manage the resources and infrastructure of cites is increasing. The utilization of the Internet of Things (IoT) in smart cities improves the quality of life by sensing, computing, and networking. Backscatter communication has high potential for commercial usage in cities, while it also has certain constraints from a practical implementation point of view. This article evaluated the coverage and outage performance of a practical cellular carrier based bi-static backscatter communication system through simulations. The simulation environments included both indoor and outdoor spaces and four different sub-GHz frequencies were applied in a real world urban city environment. For the considered system and homogeneous distribution of BDs, the 3D ray tracing simulations showed that, in an outdoor environment, the mean received power levels at BDs are around –33 dBm, –42 dBm, –45.5 dBm, and –48 dBm at 200 MHz, 500 MHz, 700 MHz, and 900 MHz, respectively. Whereas, while considering modern building infrastructure and indoor backscatter devices, the mean received power of –54.5 dBm, –58.5 dBm, and –61 dBm at 200 MHz, 500 MHz, 700 MHz, and 900 MHz, respectively, were found. The simulation results show that, at 200 MHz, the received power levels at the BDs in an outdoor environment are good enough for the purpose of energy harvesting for LoRa backscatter technology, as the power consumption for LoRa backscatter device is only 9.25 μW. However, already at 500 MHz, the time that is required for harvesting the energy is quite high for a practical backscatter communication system. It was found that, for the given scenario and setup in an outdoor environment, the mean time that is required for harvesting the energy with 40% energy harvesting efficiency is around 86 s and 880 s at 200 MHz and 500 MHz, respectively. It is emphasized that energy harvesting time can be improved by developing BDs with better energy harvesting efficiency, and by integrating the components in the BDs which can harvest the energy from sources other than incoming RF signal. We have used a hybrid approach of ray tracing and ITU-R D2D model for computing the received signal power at the reader in an outdoor environment, and for indoor receiver a hybrid approach of RT and 3GPP indoor hotspot model was used. Presented results show that 200 MHz band is well-suited for IoT and machine type communication in an urban environment for long and medium range distances in both the indoor and outdoor environment for both LoRa backscatter and NB-IoT technology. However, the simulation results also indicate that the considered bi-static BC system with NB-IoT BDs becomes coverage limited at 500 MHz in the case of indoor BDs even with 40 m or larger BD-reader separation. The coverage outage ratio of zero percent was achieved with LoRa backscatter technology in both an indoor and outdoor environment at 200 MHz for BD-reader separation up to 60 m. Whereas, at 200 MHz, in an indoor environment for NB-IoT technology, the outage ratio was found around 1.4% for 60 m BD-reader separation. The acquired results highlight the importance and potential of using lower frequency band i.e., 200 MHz and 500 MHz for IoT and backscatter communication type of services.

## Figures and Tables

**Figure 1 sensors-21-02663-f001:**
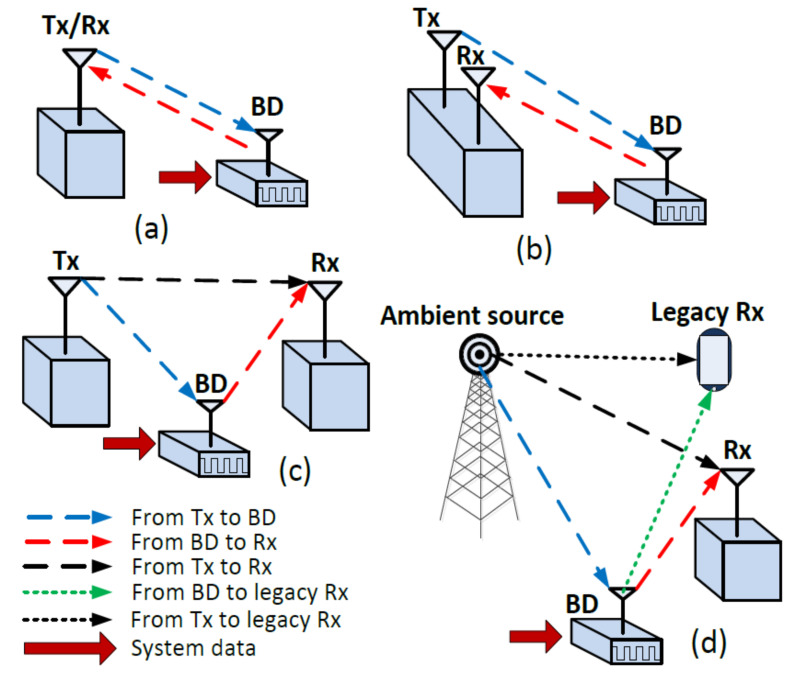
Traditional and ambient backscatter configurations, (**a**) monostatic, (**b**) bi-static co-located, (**c**) bi-static dislocated, and (**d**) ambient.

**Figure 2 sensors-21-02663-f002:**
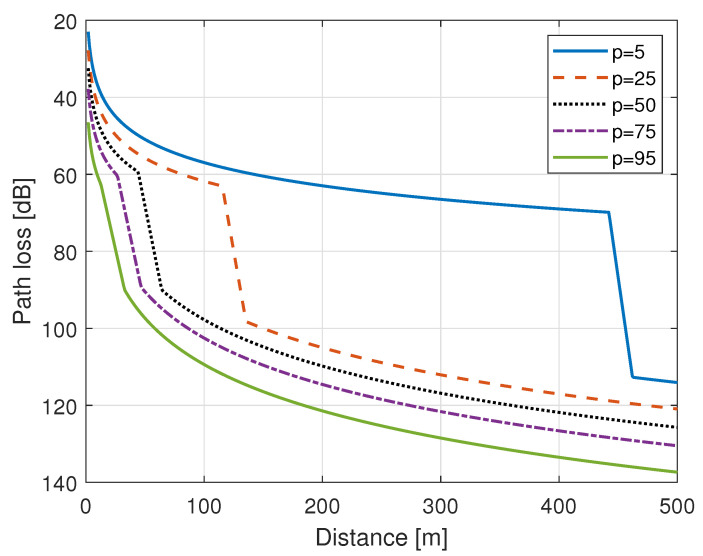
Path loss for outdoor device-to-device (D2D) communication with different location percentage.

**Figure 3 sensors-21-02663-f003:**
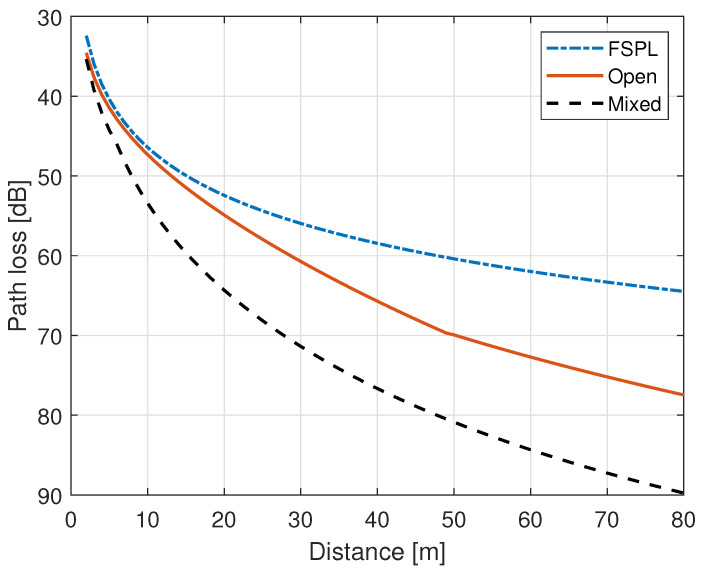
Path loss with FSPL and 3GPP InH model at 500 MHz.

**Figure 4 sensors-21-02663-f004:**
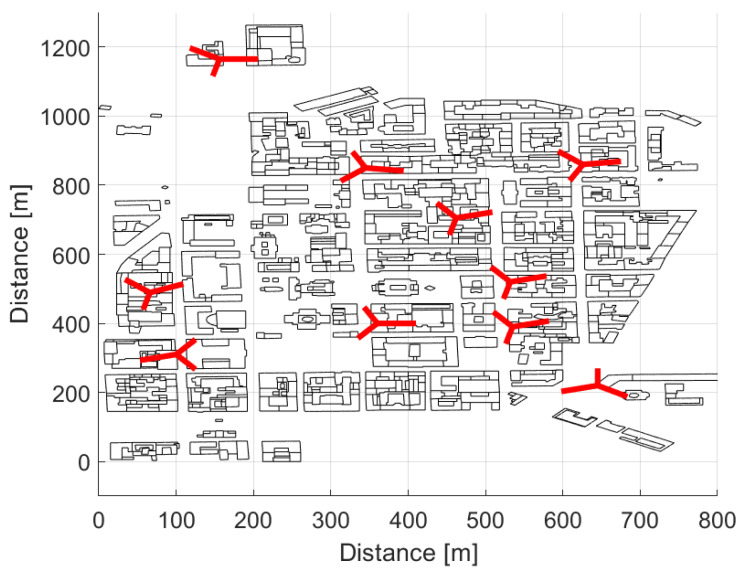
Area under consideration with site locations and cell azimuths.

**Figure 5 sensors-21-02663-f005:**
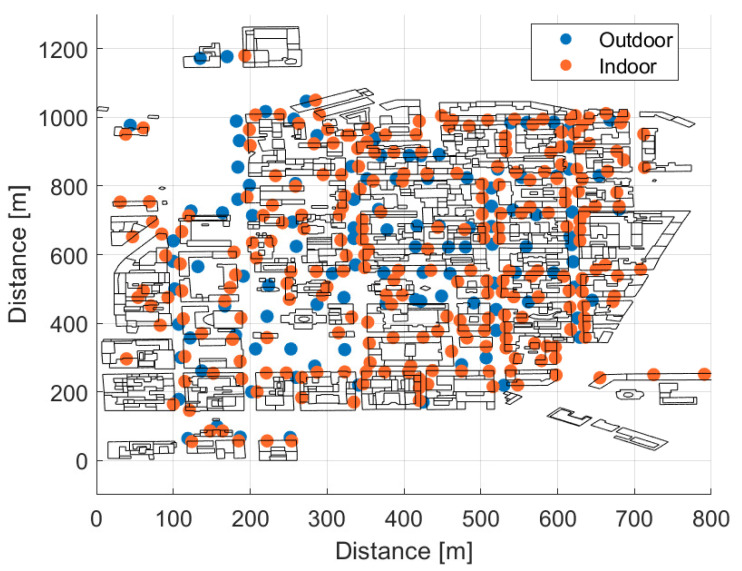
Distribution of backscatter devices.

**Figure 6 sensors-21-02663-f006:**
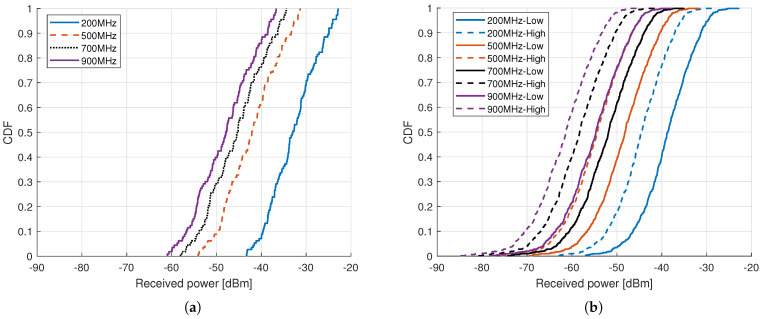
CDF of received power at backscatter device for different frequencies, (**a**) Outdoor, and (**b**) Indoor.

**Figure 7 sensors-21-02663-f007:**
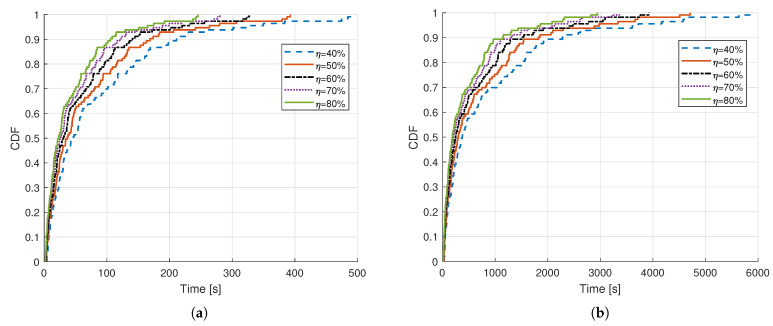
CDF of time required for energy harvesting at outdoor LoRa backscatter device with different energy harvest efficiencies at, (**a**) 200 MHz, (**b**) 500 MHz.

**Figure 8 sensors-21-02663-f008:**
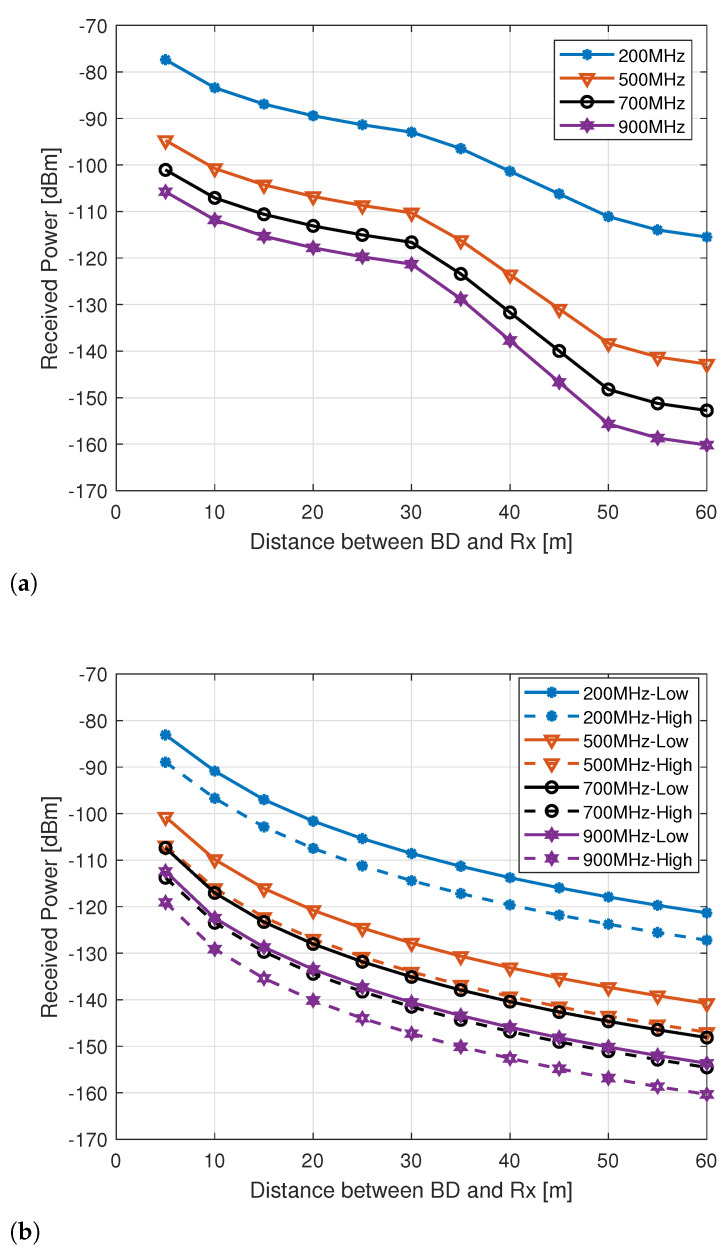
Mean received power with different BD-Rx distance at the receiver for different frequencies, (**a**) Outdoor, and (**b**) Indoor.

**Figure 9 sensors-21-02663-f009:**
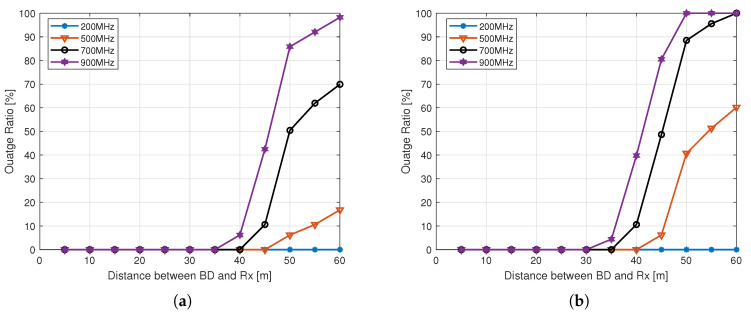
Outage ratio, (**a**) Outdoor LoRa, (**b**) Outdoor NB-IoT, (**c**) Outdoor BLE 5, (**d**) Indoor LoRa, (**e**) Indoor NB-IoT, and (**f**) Indoor BLE 5.

**Table 1 sensors-21-02663-t001:** Distribution of backscatter devices.

	Number	Percentage
Total BDs	750	100
Outdoor	113	15.06
Indoor	667	84.94
Ground floor	243	32.40
Second floor	232	30.93
Fifth floor	138	18.4
Seventh floor	24	3.2

**Table 2 sensors-21-02663-t002:** Extended 3GPP antenna model parameters.

${\mathrm{HPBW}}_{H}$ ${[}^{\circ }]$	${\mathrm{HPBW}}_{V}$ ${[}^{\circ }]$	${\mathrm{FBR}}_{H}$ [dB]	${\mathrm{SLL}}_{V}$ [dB]	$A_{\mathrm{Max}}$ [dBi]
65	7	30	18	17.7

**Table 3 sensors-21-02663-t003:** General simulation parameters.

Parameters	Unit	Value
Frequency	MHz	200/500/700/900
Tx Power	dBm	43
Antenna height	m	30
Max Tx antenna gain	dBi	17.7
BD antenna gain	dBi	0
Rx antenna gain	dBi	0
Fast fading margin (Forward)	dB	10
Fast fading margin (Backscatter)	dB	0
Slow fading margin (Forward)	dB	7
Slow fading margin (Backscatter)	dB	variable
Polarization mismatch loss	dB	3
Modulation loss at BD	dB	6

**Table 4 sensors-21-02663-t004:** Sensitivity level and maximum supported data rates for different IoT technologies.

Technology	Sensitivity	Max Data Rate	Power Consumption	Reference
BLE 5	–97 dBm	2 Mbps	20 mW	[16]
NB-IoT	–141 dBm	204.8 kbps	543 mW	[14]
LoRa backscatter	–149 dBm	37.5 kbps	9.25 uW	[55]

## Data Availability

The datasets used and/or analysed during the current study are available from the corresponding author on reasonable request.

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
