# Peer review of "Ultra-Low-Power Wide Range Backscatter Communication Using Cellular Generated Carrierâ€"

_sensors, 2021, doi:10.3390/s21082663_

Round 1
Reviewer 1 Report
The authors present a comprehensive coverage study of a practical cellular carrier-based BC system for indoor and outdoor scenarios, using as testbed scenario a downtown area of a Helsinki city. The paper is sound, well-structured and written so I recommend the publication in its present form.
Author Response
Reviewer 1 comments and their replies:
- The authors present a comprehensive coverage study of a practical cellular carrier-based BC system for indoor and outdoor scenarios, using as test bed scenario a downtown area of a Helsinki city. The paper is sound, well-structured and written so I recommend the publication in its present form.
Answer: Thanks for the appreciating words and for recommending the manuscript for publication.
Reviewer 2 Report
The submitted manuscript sensors-1109135 presents a coverage study of a cellular carrier-based backscatter communication system for indoor and outdoor scenarios. In general, the manuscript is organized and written well. Several comments are given as follows.
The authors may rewrite the key contributions to highlight the discovery of the study in this paper. The bullets from Line 70 to Line 83 only summarize what have been evaluated in the study. The current bullets can be considered as outlines rather than contributions. To emphasize the key contributions, some specific findings and obtained knowledge of those evaluations can be presented here.
It looks like both Section 2 and Section 3 introduce the background and related work in literature without providing any novelties of this paper. Consequently, Section 4 introduces the simulation settings of this work. The writing style in this way is more like a technical report rather than an article in a journal. The authors are suggested to modify the writing style, particularly of Section 4. For example, the authors may present it as a new developed Case Study Design to achieve the research goal, rather than leading readers to feel like it is a simulation testing of those past work that is introduced in Section 3.
Since only simulations are conducted in this study, the authors may provide discussions about some possible issues or differences between simulations and real-world scenarios when implementing a cellular carrier-based backscatter communication system.
Author Response
Reviewer 2 comments and their replies:
- The submitted manuscript sensors-1109135 presents a coverage study of a cellular carrier-based backscatter communication system for indoor and outdoor scenarios. In general, the manuscript is organized and written well. Several comments are given as follows.
Answer: Thanks for the encouraging and appreciating words. Authors would definitely like to consider your comments in order to improve the quality of the manuscript.
- The authors may rewrite the key contributions to highlight the discovery of the study in this paper. The bullets from Line 70 to Line 83 only summarize what have been evaluated in the study. The current bullets can be considered as outlines rather than contributions. To emphasize the key contributions, some specific findings and obtained knowledge of those evaluations can be presented here.
Answer: As suggested by the reviewer, now authors have also highlighted few of the main findings of the work. As mentioned by the reviewer, earlier only the research items were listed, however, now for each listed item we have also highlight the key findings and presented them as the contribution of this research work. We have created a new subsection 1.1 Targets and key contributions, and there specific findings and key contributions are highlighted.
- It looks like both Section 2 and Section 3 introduce the background and related work in literature without providing any novelties of this paper. Consequently, Section 4 introduces the simulation settings of this work. The writing style in this way is more like a technical report rather than an article in a journal. The authors are suggested to modify the writing style, particularly of Section 4. For example, the authors may present it as a new developed Case Study Design to achieve the research goal, rather than leading readers to feel like it is a simulation testing of those past work that is introduced in Section 3.
Answer: Here, the Section 2. Background provides the background theory about LPWAN, and briefly introduced the reader with the existing different LPWAN technologies. This research work mainly deals with AmBC, therefore the difference between traditional and ambient BC is highlighted, and few of the applications of AmBC are listed, and related work is shown in Section 2. The Section 3. Propagation models provides the technical detail of the propagation models used for the research work of this manuscript. The key contributions and the novelty of this paper is highlighted in the last part of the Section I. Introduction. The Section 4. is now renamed as “System design and simulation configuration”, and there subsection 4.1 provides the description of the system considered for a case study, and gives detail about the environment of the system and the parameters of the simulations. Additional models and their parameters considered for the simulations are also presented in Section 4.
- Since only simulations are conducted in this study, the authors may provide discussions about some possible issues or differences between simulations and real-world scenarios when implementing a cellular carrier-based backscatter communication system.
Answer: Yes, we have only provided the simulation results in this manuscript, but we would like to emphasize on the fact that the simulation environment and the simulation parameters were selected as realistic as possible, and a well known and widely accepted models are used to acquire the simulation results. We have tried our best to select the parameter values of simulations based on some references. However, those parameter values can always vary in real life and will consequently affect the results.
Reviewer 3 Report
The article "ULTRA LOW POWER WIDE RANGE BACKSCATTER COMMUNICATION USING CELLULAR GENERATED CARRIER" has a relevant topic for research.
However, some points are not clear and need to be better justified and/or detailed:
It is not clear why the four frequencies investigated, the reason for the choice, advantages, and disadvantages for the backscattering technique (BC).
Why not bigger, smaller or intercalary potentials? size and amount of data transmitted are also unclear.
In line 289, the authors also describe that they do not consider details of the recipients, but if there is no minimum deepening what is the scientific contribution?
Even though the work is based on simulations, only.
Line 448 describes that the simulation is based on a real scenario, but would these scenarios only cover the four frequencies? I am not convinced that it would be enough.
Is there a similar technique or work published recently (last 2 years) that shows this?
I suggest these two works that present solutions implemented in IoT scenarios using communication protocols.
https://doi.org/10.3390/electronics9091430
In addition to using current scenarios, they present implementations with techniques implemented in sensors and real devices.
Is an interruption rate as described in the conclusions of 1.4% sufficient? for what amount of data received and sent?
I also suggest a list of symbols for the mathematical equations used along with the list of abbreviations.
References
Altogether there are 53, with only 16 recent (last 3 years).
Several references do not have complete information, it is imperative that they adjust. DOI or ISSN, and links with access date.
Between 2020 and 2021 there is no recent work involving the researched state of the art?
I suggest you review the literature for the final version.
I hope I have contributed to improving your work.
Author Response
Reviewer 3 comments and their replies:
- The article "ULTRA LOW POWER WIDE RANGE BACKSCATTER COMMUNICATION USING CELLULAR GENERATED CARRIER" has a relevant topic for research.
However, some points are not clear and need to be better justified and/or detailed:
- It is not clear why the four frequencies investigated, the reason for the choice, advantages, and disadvantages for the backscattering technique (BC).
Answer: The target of this study is to cover the existing cellular frequencies and the candidate frequency bands for IoT networks. It must be noted here that 900 MHz band is mainly used by cellular systems, and a large number of existing cellular sites use this band for providing basic cellular coverage. In Europe, the 700 MHz spectrum is available for digital television broadcasting, and is also considered as a potential band for deploying 5G of cellular system. Currently, in some European countries there is a debate going on about the usage of 200 MHz and different proposals are given, and the use of IoT at 200 MHz is one of them. The 500 MHz is considered as an intermediate band between 200 and 700 MHz band. Therefore, for the research work of article, the simulations are performed at four different frequencies i.e., at 200 MHz, 500 MHz, 700 MHz, and 900 MHz. (This information is added in the section 4.4. General system parameters)
The main advantage of the AmBC is that it utilizes the abundant existing radio frequency signal (e.g., cellular network, WLAN, TV and radio broadcast) for transmitting information from the BD to the Rx and also for harvesting the energy at the BD. More specifically, ambient BD needs no dedicated energy provider, and it reduces the cost of installation and maintenance. Also, ambient backscatter enables D2D communication, different from the traditional backscatter that must communicate exclusively with a reader, without sensing and communicating with other nearby BDs. (This information is available in Section 2.2.2 Ambient)
- Why not bigger, smaller or intercalary potentials? size and amount of data transmitted are also unclear.
In line 289, the authors also describe that they do not consider details of the recipients, but if there is no minimum deepening what is the scientific contribution?
Answer: In line 289 “we only consider the sensitivity level of the receiver, and do not go into the detail of receiver design” by this statement we mean that we have used the sensitivity levels and the receiver design criteria of different existing technologies i.e., LoRa backscatter, NB-IoT, and BLE5 for evaluating the coverage analysis and the feasibility of a considered system in our case study. Those devices already exist and we used the technology specification i.e., sensitivity level, maximum supported data rate, and power consumption of those devices in our study, and they are now given in Table 4.
(Line 289 in previous manuscript is now modified with clarification, and is now written as the last line of Section 2.5. Related work) “In this work, we have used the sensitivity levels and the receiver design criteria of different existing technologies i.e., LoRa backscatter, NB-IoT, and BLE5 for evaluating the coverage and the feasibility of a cellular generated carrier based backscatter communication system in our case study.”
- Even though the work is based on simulations, only. Line 448 describes that the simulation is based on a real scenario, but would these scenarios only cover the four frequencies? I am not convinced that it would be enough. Is there a similar technique or work published recently (last 2 years) that shows this?
Answer: In Line 448, we meant that instead of hypothetical regular Manhattan type building grid with equal building height or simplistic green field plan we have considered a 3D building map of real city (Helsinki, city of Finland) for three dimensional ray tracing. Our target is to evaluate the feasibility of existing cellular networks in implementing a cellular generated carrier based backscatter communication system and that’s why we have considered those four frequencies for our study. Similar study can be done for other frequencies; however, we have focused on those four considered frequencies. Again, to the best of our knowledge there is no study available in literature in which they have done such a comprehensive campaign of simulations using 3D ray tracing and other analytical model for evaluating the feasibility of existing cellular generated carried based backscatter communication system.
- I suggest these two works that present solutions implemented in IoT scenarios using communication protocols.
https://doi.org/10.3390/electronics9091430
- In addition to using current scenarios, they present implementations with techniques implemented in sensors and real devices.
Answer: Sorry, you talked about two references in your comment but you provided only one reference link. The article “Multi-Agent Approach Using LoRaWAN Devices: An Airport Case Study” deals with the application of LoRaWAN devices in an airport washroom case study. They used LoRa devices for couple of applications as given in the reference article, and their target was to show the applications of LoRaWAN technology. However, our research target is different from their research work, as our target is to show the viability of cellular generated carrier based backscatter communication system in an urban city environment, and for that purpose we considered the required sensitivity level of different IoT technologies as an input (requirement) for the coverage analysis. We have not developed our own devices, rather we have used the specifications of already existing devices for our analysis.
- Is an interruption rate as described in the conclusions of 1.4% sufficient? for what amount of data received and sent?
Answer: In our research work, “The outage ratio is defined as the ratio of the locations with received power below a receiver sensitivity threshold to the total number of receiver points considered in the area.” In this article, we have shown the outage ratio with respect to different IoT technologies and for different for backscatter link distances. As far as your question is concerned that is 1.4% outage ratio is sufficient or not ? We leave it on to the implementation guys that what outage ratio is acceptable to them. For example in a critical mission cases 1.4% outage is considered as high, however in other applications it can be considered at an acceptable level. The amount of maximum data transfer for different communication technologies is given in Table 4 now, as different technologies support different data rates.
- I also suggest a list of symbols for the mathematical equations used along with the list of abbreviations.
Answer: Yes, according to your suggestion, the list of symbols of mathematical equations has been added after abbreviations list, and the description of each symbol is also added there. I hope it will be helpful for the reader as well.
- Altogether there are 53, with only 16 recent (last 3 years). Several references do not have complete information, it is imperative that they adjust. DOI or ISSN, and links with access date. Between 2020 and 2021 there is no recent work involving the researched state of the art? I suggest you review the literature for the final version.
Answer: Yes, according to your suggestion, the information of all references such as DOIs or ISSNs has been checked and rectified. There are technical white papers and standard specifications in reference which has no specific DOI to provide. Besides, updated references have been added, and now there are 9 references between year 2020 and 2021 (2, 3, 24, 25, 36, 37, 38, 40, 44). Few old references are mainly for ray tracing. I hope you will be satisfied with the current ratio of old and new references.
Reviewer 4 Report
This paper evaluates the coverage and outage performance of a practical cellular carrier-based backscatter communication system for indoor and outdoor scenarios.
The paper shows interesting simulation results; however, the authors should improve the technical quality of the paper by following these recommendations:
- The authors should remove all the well-known details (mainly in sections 2 and 3) and give more attention to explain the new contribution.
- The theoretical part of the paper should be more detailed.
- The authors should explain why only the sensitivity level of the receivers is considered.
- Why only two simulation cases have been considered?
- How it is possible to integrate other cases or scenarios?
- The simulation methodology in section 4 needs to be deeply and technically described.
- The source code of the implemented propagation and other models should be explained.
- The choice of the simulation parameters given in Table 3 should be justified.
- what is the interest to use various frequencies and various technologies?
- The acquired simulation results should be supported by experimental validation in a real scenario.
- The validation of the proposed approach should be given, and should be done by comparisons between the measurements, the obtained simulation results and the state of the art.
Author Response
Reviewer 4 comments and their replies:
The paper shows interesting simulation results; however, the authors should improve the technical quality of the paper by following these recommendations:
- The authors should remove all the well-known details (mainly in sections 2 and 3) and give more attention to explain the new contribution.
Answer: Section 2. Background mainly provides the background theory about LPWAN, and now in updated manuscript as recommended we have removed extra information from all subsections of Section 2 and that makes Section 2 more concise compared to before. The Section 3 Propagation models provides the technical detail of the propagation models used for the research work of this manuscript, and we have tried to provide all necessary details of the models in Section 3 which we have used in our analysis. Now, only directly relevant and required information is given in Section 3. The main targets, key contributions and the novelty of this article are highlighted in the last part of the Section I. Introduction. Please check updated Section I for more detail.
- The theoretical part of the paper should be more detailed.
Answer: Basically, Section 2 and Section 3 deals with the theoretical part of this research work, and now Section 4 is renamed as “System design and simulation configuration”, and there subsection 4.1 provides the description of the system considered for a case study, and gives detail about the environment of the system and the parameters of the simulations. Now we have added pseudocode of three propagation models i.e., ray tracing, ITU outdoor D2D model and for 3GPP Indoor hot spot model. Please feel free to let us know which topic needs more detail.
- The authors should explain why only the sensitivity level of the receivers is considered.
Answer: In section 1.2 Targets and key contribution, it is stated “In particular, we evaluate the outage ratio of a BC system based on the receiver sensitivity level of different low-power wide area technologies through comprehensive set of simulations in an urban city environment.”. At first we computed the received power level at backscatter devices and at reader at different frequencies in an outdoor and indoor environment, and then with respect to the sensitivity level of the receiver of different IoT technologies we found the outage ratio. Here, I would like to remind that in the manuscript “The outage ratio is defined as the ratio of the locations with received power below a receiver sensitivity threshold to the total number of receiver points considered in the area.”
For energy harvesting time, the received power level and the power consumption of LoRa backscatter device is used an in input.
- Why only two simulation cases have been considered?
Answer: In order to keep separate indoor and outdoor analysis, the simulation scenarios are mainly divided into two broad classes i.e., 1) Outdoor BD with outdoor reader, and 2) Indoor BD with indoor reader. Also, we have used two separate large scale propagation models for outdoor and indoor i.e., ITU outdoor D2D model and 3GPP Indoor hotspot model for outdoor and indoor BD-reader link, respectively. A hybrid case of outdoor and indoor BDs and reader will make the scenario complex, and the direct use of earlier mentioned propagation models is not possible. Therefore, in order to keep analysis simple, only two broad classes of cases are considered.
- How it is possible to integrate other cases or scenarios?
Answer: There can be two more cases i.e., outdoor BD with indoor reader and indoor BD with outdoor reader. It must be noted here that for forward link i.e. between the transmitter and BD, we have utilized ray tracing tool for finding the propagation path between the Tx and the BD, and that BPL is taken into account for computing the received power at indoor BD. We can still use that information for new cases, and then backscatter link i.e., between BD and the reader will have two parts (one in outdoor and second in indoor environment). We need to use ITU outdoor D2D model for the outdoor part and 3GPP Indoor hotspot model for the indoor part, and again need to consider the building penetration loss for the second time.
- The simulation methodology in section 4 needs to be deeply and technically described.
Answer: Section 4.1 discusses in detail about the environment, clearly showing the layout of the buildings, location of the transmitters, locations of the backscatter devices. The Section 4.3 Simulation tool, Models and parameters provides the detail about the simulation tool used, and presents additional models used in our simulations, and finally Section 4.4 provides a complete list of general parameters used in our simulations. If anything is not clear, then please feel free to let us know.
- The source code of the implemented propagation and other models should be explained.
Answer: We fully agree with your suggestion, and now we have added pseudocode of three propagation models i.e., ray tracing, ITU outdoor D2D model and for 3GPP Indoor hot spot model. I hope it will help the reader in better understanding the working principle of these models.
- The choice of the simulation parameters given in Table 3 should be justified.
Answer: Now, the values of the simulation parameters used in Table 3 and Table 4 are justified and proper reference is given for each used value.
- what is the interest to use various frequencies and various technologies?
Answer: As mentioned earlier that “the main target of this study is to cover few of the existing cellular frequencies and some candidate frequency bands for IoT networks. It must be noted here that 900 MHz band is mainly used by cellular systems, and a large number of existing cellular sites use this band for providing basic cellular coverage. In Europe, the 700 MHz spectrum is available for digital television broadcasting, and is also considered as a potential band for deploying 5G of cellular system. Currently, in some European countries there is a debate going on about the usage of 200 MHz and different proposals are given, and the use of IoT at 200 MHz is one of them. The 500 MHz is considered as an intermediate band between 200 and 700 MHz band. Therefore, for the research work of article, the simulations are performed at four different frequencies i.e., at 200 MHz, 500 MHz, 700 MHz, and 900 MHz.” (This information is added in the section 4.4. General system parameters)
We would like to highlight here that backscatter based LoRA [53] and BLE [55] radios have already been implemented and we have selected NB-IoT for the sake of comparison. We also would like to emphasis that these technologies can be refarmed to the studied bands with change of RF front-end, and it should not significantly impact the sensitivity level.
[53] Talla, V.; Hessar, M.; Kellogg, B.; Najafi, A.; Smith, J.R.; Gollakota, S. LoRa Backscatter: Enabling The Vision of Ubiquitous Connectivity. Proc. ACM Interact. Mob. Wearable Ubiquitous Technol.2017,1. doi:10.1145/3130970.
[55 ]Ensworth, J.F.; Reynolds, M.S. BLE-Backscatter: Ultralow-Power IoT Nodes Compatible With Bluetooth 4.0 Low Energy (BLE) Smartphones and Tablets. IEEE Transactions on Microwave Theory and Techniques 2017,65, 3360–3368. doi:10.1109/TMTT.2017.2687866.
We have computed the outage ratio with respect to three technologies i.e., LoRa backscatter, NB-IoT and BLE5, as these are the few well known IoT technologies, and have clearly different sensitivity thresholds and supported data rates, and are suitable for different types of IoT applications. Main features of three considered technologies are now listed in Table 4.
- The acquired simulation results should be supported by experimental validation in a real scenario.
The validation of the proposed approach should be given, and should be done by comparisons between the measurements, the obtained simulation results and the state of the art.
Answer: In our considered case study, we have placed 750 BDs at different outdoor and indoor positions, and in order to acquire the real measurement data of a considered cellular generated carrier based backscatter communication system, a large scale backscatter system needs to be deployed and a huge amount of resources are required, which are not currently available to us. Definitely, we would like to validate our simulation results with the measurement data, but unfortunately that is not currently possible for us. However, here we still like to emphasize on the fact that we have considered a quite realistic set of simulation parameters.
Reviewer 5 Report
The work presented in this paper is a preliminary work that aims to evaluate cellular carrier-based backscatter communications using low-power and wide-area technologies, with a focus on three prominent LPWANs, namely LoRa, NB-IoT, and BLE for indoor and outdoor applications. However, this study was only evaluated in a simulation environment, and has not been validated experimentally. Moreover, there are issues related to the reporting and validation of the main finding that, in my opinion, make it difficult to clearly assess the main contributions and compromise the article's acceptance.
The paper lacks a more profound introduction of the LPWAN technologies under analysis and the literature review can be extended to include more works and techniques that are not addressed in the current version of the document. Please consider the works and the techniques in reference [1].
The evaluation was undertaken only in simulation and no indoor or outdoor real-world data was considered. Moreover, static and dynamic node scenarios were not properly distinguished. The propagation models adopted do not seem to be aware of the possibility of having nodes in movement (i.e. dynamic nodes) and do not consider its main problems in urban environments, i.e. i.e. urban canyons, where multipath and fading situations regularly occur, and some questions arise. For example, what happens when using LoRa technology in moving nodes for a small spreading factor, and how does it impact the LoRa-based backscatter communications?
Overall, and based on the comments below, I suggest the paper to be considered for publication after a major review.
Regarding presentation:
The overall presentation of the paper needs to be improved.
The title should not be in CAPS. Moreover, the term “Ultra” seems excessive. Please clarify in the document what actually is “ultra-low-power” with relation to “low-power”, and exemplify based on evidence, and using a quantitative approach.
“Section II - Background” needs to be improved. The introduction of the technologies addressed need to be more elaborate with a focus on the relevant figures of merit that are used in the evaluation of backscatter communications.
“Section 2.5 - Related Work” should be extended to include more workers in the topic. Moreover, the introduction of the related works should be more elaborated and extended.
The quality of the images is good. However, I would recommend to the authors, in general, more care in the preparation of figures and images. Try to normalize the font sizes in all images (e.g. figure 1 has a huge font size);
Keywords should include the technologies used, for example, ”NB-IoT” and “BLE” and “LPWAN”.
The overall presentation of the paper can still be improved, and written English should be revised. Moreover, the paper has a mix between British English and US English:
- urbanisation vs urbanization
- utilisation vs. utilization
Although the simulation methodology seems appropriate, the final evaluation is only performed in a simulation environment which is a major drawback for the current work. Additional real-world data (…at least one technology addressed) would be a plus and would raise the quality of the article to a higher level. With this, a validation of the proposed approach would be possible and the contribution of this article would become more irrefutable.
References:
[1] F. Pereira, R. Correia, P. Pinho, S. I. Lopes, N. B. Carvalho, “Challenges in Resource-Constrained IoT Devices: Energy and Communication as Critical Success Factors for Future IoT Deployment”, Sensors 2020, 20, 6420, DOI: 10.3390/s20226420
Author Response
Reviewer 5 comments and their replies:
- The work presented in this paper is a preliminary work that aims to evaluate cellular carrier-based backscatter communications using low-power and wide-area technologies, with a focus on three prominent LPWANs, namely LoRa, NB-IoT, and BLE for indoor and outdoor applications. However, this study was only evaluated in a simulation environment, and has not been validated experimentally. Moreover, there are issues related to the reporting and validation of the main finding that, in my opinion, make it difficult to clearly assess the main contributions and compromise the article's acceptance.
Answer: Yes, we have only provided the simulation results in this manuscript, but we would like to emphasize on the fact that the simulation environment and the simulation parameters were selected as realistic as possible, and a well known and widely accepted models are used to acquire the simulation results. We have tried our best to select the parameter values of simulations based on given references. However, those parameter values can always vary in real life and will consequently affect the results. In a revised manuscript, we have created a new subsection 1.1 Targets and key contributions, and specific findings and key contributions are highlighted there.
- The paper lacks a more profound introduction of the LPWAN technologies under analysis and the literature review can be extended to include more works and techniques that are not addressed in the current version of the document. Please consider the works and the techniques in reference [1].
[1] F. Pereira, R. Correia, P. Pinho, S. I. Lopes, N. B. Carvalho, “Challenges in Resource-Constrained IoT Devices: Energy and Communication as Critical Success Factors for Future IoT Deployment”, Sensors 2020, 20, 6420, DOI: 10.3390/s20226420
Answer: There is a complete Section 2.1 Low power wide area network where a short background about LPWAN is given, and later in subsections few of the LPWAN technologies are introduced. Like your given reference, we have briefly introduced LoRa, SigFox, NB-IoT, BLE, Ingenu, D7AP. We are mainly targeting long range communication in our work, therefore short range IoT technologies like Zigbee, NFC, WirelessHART are not considered. The reference [1] is also cited in revised manuscript now.
- The evaluation was undertaken only in simulation and no indoor or outdoor real-world data was considered. Moreover, static and dynamic node scenarios were not properly distinguished. The propagation models adopted do not seem to be aware of the possibility of having nodes in movement (i.e. dynamic nodes) and do not consider its main problems in urban environments, i.e. i.e. urban canyons, where multipath and fading situations regularly occur, and some questions arise. For example, what happens when using LoRa technology in moving nodes for a small spreading factor, and how does it impact the LoRa-based backscatter communications?
Answer: In our considered case study, we have placed 750 static BDs at different outdoor and indoor positions, and in order to acquire the real measurement data of a considered cellular generated carrier based backscatter communication system, a large scale backscatter system needs to be deployed and a huge amount of resources are required, which are not currently available to us. Definitely, we would like to validate our simulation results with the measurement data, but unfortunately that is not currently possible for us. However, here we still like to emphasize on the fact that we have considered a quite realistic set of simulation parameters and utilized a 3D building map of Helsinki city for ray tracing simulation part.
Now, it is explicitly mentioned in the manuscript that we have considered only “Static” BDs and the mobility of devices are not considered in our case study. As far as impact of multipath and street canyon propagation is concerned, I would like to remind here that we have used 3D ray tracing simulation for forward link computation and that utilizes 3D building data of the environment and finds the multipaths with given number of reflections and diffraction, which means ray tracing simulation already consider the impact of multipath propagation and street canyons for the forward link.
- Overall, and based on the comments below, I suggest the paper to be considered for publication after a major review.
- The overall presentation of the paper needs to be improved.
Answer: We have made several modifications in our revised manuscript and have added the pseudocodes of three propagation models as well for better understanding those models. The key targets and contributions are highlighted. Some more new references have been added, and few old ones are deleted. Similarly, well-known information was deleted to make a better flow of the manuscript. I hope, this time you will find it better.
- The title should not be in CAPS. Moreover, the term “Ultra” seems excessive. Please clarify in the document what actually is “ultra-low-power” with relation to “low-power”, and exemplify based on evidence, and using a quantitative approach.
Answer: Thanks for the suggestion; the title is now changed to “Ultra-Low-Power Wide Range Backscatter Communication Using Cellular Generated Carrier”. In manuscript it is mention “Ultra-low power operation enables the long-term stability, and makes it easy to maintain IoT devices for longer period of time, and it decreases the cost, energy consumption and devices maintenance.”
- “Section II - Background” needs to be improved. The introduction of the technologies addressed need to be more elaborate with a focus on the relevant figures of merit that are used in the evaluation of backscatter communications.
Answer: “Section II - Background” and “Section III Propagation Models” have been modified.
- “Section 2.5 - Related Work” should be extended to include more workers in the topic. Moreover, the introduction of the related works should be more elaborated and extended.
Answer: “Section 2.5 - Related Work” has been extended and few more references have been added in the revised manuscript.
- The quality of the images is good. However, I would recommend to the authors, in general, more care in the preparation of figures and images. Try to normalize the font sizes in all images (e.g. figure 1 has a huge font size);
Answer: The font size of Figure 1 is purposely set large so that Figure along with text and legends are clearly visible. Rest, all of the results figures were made using MATLAB, same and clear font size is used in all figures made by MATLAB.
- Keywords should include the technologies used, for example, ”NB-IoT” and “BLE” and “LPWAN”.
Answer: Keywords have been added as recommended.
- The overall presentation of the paper can still be improved, and written English should be revised. Moreover, the paper has a mix between British English and US English:
- urbanisation vs urbanization
- utilisation vs. utilization
Answer: The overall paper is reviewed for language, spelling and grammar mistakes. Other than that a mix of British and US English is rectified as recommended.
- Although the simulation methodology seems appropriate, the final evaluation is only performed in a simulation environment which is a major drawback for the current work. Additional real-world data (…at least one technology addressed) would be a plus and would raise the quality of the article to a higher level. With this, a validation of the proposed approach would be possible and the contribution of this article would become more irrefutable.
Answer: Yes, we agree with you that the validation of simulation results with the measurement data will definitely improve the value of our work and we would like to carry such measurements, but unfortunately that is not currently possible for us.
Round 2
Reviewer 2 Report
The authors have addressed all the comments and concerns in my previous review report. The manuscript is suggested to be accepted.
Reviewer 4 Report
The authors have satisfactorily addressed my concerns.
The authors have done several efforts to improve the technical quality of the paper. For this reason, the manuscript can be accepted for publication.
Reviewer 5 Report
All my questions have been answered, and my suggestions have been addressed satisfactorily. Moreover, taking into account that additional information based on the suggestions of other reviewers have also been included, the overall quality of the paper has been improved and given the quality of the presented work and its improvements after revision I recommend this paper to be accepted.